# Intensive Sleep Re-Training: From Bench to Bedside

**DOI:** 10.3390/brainsci7040033

**Published:** 2017-03-27

**Authors:** Leon Lack, Hannah Scott, Gorica Micic, Nicole Lovato

**Affiliations:** 1School of Psychology, Flinders University of South Australia, GPO Box 2100, Adelaide 5001, Australia; scot0307@uni.flinders.edu.au (H.S.); gorica.micic@flinders.edu.au (G.M.); nicole.lovato@flinders.edu.au (N.L.); 2School of Medicine, Adelaide Institute for Sleep Health: A Flinders Centre of Research Excellence, Flinders University, Adelaide 5001, Australia

**Keywords:** insomnia, cognitive behavioral therapy, CBT-I cognitive therapy, intensive sleep re-training, sleep deprivation

## Abstract

Intensive sleep re-training is a promising new therapy for chronic insomnia. Therapy is completed over a 24-h period during a state of sleep deprivation. Improvements of sleep and daytime impairments are comparable to the use of stimulus control therapy but with the advantage of a rapid reversal of the insomnia. The initial studies have been laboratory based and not readily accessible to the patient population. However, new smart phone technology, using a behavioral response to external stimuli as a measure of sleep/wake state instead of EEG determination of sleep, has made this new therapy readily available. Technological improvements are still being made allowing the therapy to provide further improvements in the effectiveness of Intensive Sleep Re-training.

## 1. Introduction

Chronic insomnia is not a trivial disorder in our society. Ongoing frequent difficulties initiating sleep and/or getting back to sleep after awakenings during the night may be troublesome at best. However, they often lead to underlying psychophysiological responses such as anxiety, frustration, anger, and a full-fledged sympathetic nervous system “fight-or-flight” response. These reactions not only prolong wakefulness, but if they occur repeatedly in the same conditions (e.g., in bed, at night, attempting sleep, etc.), the mere conditions can become triggers for these psychophysiological reactions. These reactions can become chronic and indicate the development of conditioned or “learned” insomnia through simple associations of the bed environment and the psychophysiological reactions by well-known classical (Pavlovian) conditioning [1].

The aspects of chronic insomnia (duration >3 months) that are also essential to define the disorder are the daytime impairments associated with the sleeping difficulties. These aspects tend to be the most troublesome and most likely to motivate the sufferer to seek treatment [2]. They include feelings of mental and physical fatigue, cognitive and memory impairments, emotional distress, and mild depression [3]. In fact, chronic insomnia is a high risk factor for the development of severe depression in adolescents and adults [4,5].

Chronic insomnia is not only a burden experienced by the individual, it is a social and economic burden for society [6,7]. One study found the additional annual per person cost of chronic insomnia was over $4000, mainly due to lost work and productivity [8]. The prevalence of this disorder from surveys around the world ranges from 4%–6% [9]. Calculating the financial cost to society from the individual burden and prevalence rates reveal that chronic insomnia is a significant societal burden (e.g., ~$64 billion/year in the USA).

A common strategy for sufferers attempting to treat their insomnia is to try a whole range of over-the-counter substances that promise better sleep, but this strategy provides little and often dubious long-term benefit [10]. Many eventually visit a medical practitioner from whom they are very likely to obtain a hypnotic drug prescription [11]. Unfortunately, hypnotic drugs only provide symptomatic relief and do not cure the insomnia because they target symptoms and not the predisposing/perpetuating causes of the insomnia. They also have significant side effects, are duration-limited in relieving symptoms, and are associated with risk of drug dependency [12].

The organization of this review paper will focus on the effective non-drug treatment of insomnia with cognitive/behavior therapy, some of its shortcomings, the promising results from a novel therapy (intensive sleep re-training), and its possible translation from laboratory administration to home-based treatment with new technology. The article will conclude with a description of further technological developments for administering this novel therapy and the opportunities it will provide for carrying out extensive variations of its procedure to further improve the effectiveness of insomnia treatment. The ultimate aim is to provide a highly effective, inexpensive, and readily accessible treatment of insomnia. 

## 2. Cognitive/Behavior Therapy for Insomnia (CBT-I)

CBT-I is a multi-component non-drug treatment package that includes a range of procedures and interactions between the therapist and patient. Components include education about sleep, sleep hygiene suggestions (e.g., regularizing time in bed, reducing caffeine consumption, etc.), cognitive therapy to address maladaptive beliefs about sleep, relaxation practices, and behavioral therapies. CBT-I has been shown to be equally effective to pharmacotherapy in the short term and more durable following treatment [13]. Treatment is designed to address the specific causes of the patient’s insomnia, thus reducing their particular sleeping difficulties and daytime impairment symptoms. The components emphasized for treatment may be tailored to the patient’s insomnia phenotype with the prospect of maximizing treatment effectiveness. For example, an exaggerated fear of losing any sleep would be appropriately treated with cognitive therapy.

In any case, behavior therapies are the mainstay of CBT-I, providing the most rapid improvement for the majority of insomnia sufferers [14]. These therapies address the maladaptive learning or association process mentioned above that results in hyperarousal while attempting sleep. The two most applied behavior therapies are stimulus control therapy (SCT) [15] and bed period or sleep restriction therapy (SRT) [16].

SCT is the most widely studied and endorsed single component treatment method and has become the “gold standard” for the testing of new interventions [1].

The instructions are as follows:Do not have a pre-determined bed time, go to bed only when sleepy;Get out of bed if not asleep within 15 min;Repeat #1 and #2 until a rapid sleep onset occurs;Maintain the same wake-up time regardless of sleep length; andDo not nap during the day.

By increasing the instances of falling asleep quickly in one’s bed, these instructions aim to re-associate the bed environment with sleep and reverse the conditioned insomnia. Adjunct suggestions include eliminating other behaviors incompatible with sleep (e.g., using electronic screen devices, eating, arguments with bed partner, worrying about losing sleep) from the bedroom environment. Although SCT is effective for both sleep onset and sleep maintenance insomnia, its focus on initial sleep onset indicates its use particularly when initial sleep onset problems are predominant.

If sleep maintenance problems are predominant (i.e., excessive periods of wakefulness during the night and excessive amount of time spent in bed), SRT is a logical choice. As its name implies, the therapy consists of restricting time in bed (e.g., later bedtime and/or earlier out of bed in the morning) to equate with the average amount of reported total sleep time during one week of sleep diary. However, patients are not restricted to less than 5 h of time in bed per night, even if the total sleep time indicated on the sleep diary is less than that. Sleep efficiency (% of time in bed asleep) and excessive sleepiness are monitored weekly before and during treatment. Over 1–2 weeks of SRT, sleep typically becomes more consolidated and sleep efficiency increases. When it rises above 85% and when sleepiness, usually apparent in the hour or two before bedtime, becomes difficult to resist, time in bed can then be extended in small increments (e.g., 15 min per week) as long as a sleep efficiency >85% is maintained. 

After the initial period of sleep restriction, the subsequent extension of time in bed is typically associated with reported improvements in daytime functioning and feelings of greater energy [17]. Sleep restriction therapy has moderate guideline support for its efficacy and recent strong clinical research support [17,18]. The assumed mechanism of action is the eventual conditioning of a rapid sleep onset and/or return to sleep, reducing the learned hyperarousal of chronic insomnia.

## 3. Challenges to the Implementation of Behavior Therapies

Although the two behavior therapies have somewhat different instructions, they share common features that directly address the major perpetuating factors of chronic insomnia. Both therapies reduce time in bed and prohibit napping. This effectively reduces total sleep time over the first few weeks of therapy, thereby increasing homeostatic sleep drive (or sleep pressure) and reducing both chronic hyperarousal and time awake in bed. Over time, the bedroom environment and the intention of falling asleep become re-associated with rapid sleep onsets, thus reversing the previously learned insomnia response. 

Although randomized controlled trials have provided strong empirical support for the efficacy of behavior therapies, their effectiveness in clinical practice is not well documented. Baillargeon et al. [19] showed that SCT could be used effectively by family physicians but noted that it was most useful for highly motivated patients. The therapies present some challenges. Their administration is typically associated with a lag in treatment response (3–4 weeks), some early treatment daytime sleepiness, and difficulties with treatment compliance [20]. Compliance is necessary for successful treatment but is often suboptimal [21], very likely arising from the difficulty in changing lifestyle habits at home and the sleep loss over several weeks of the treatment. Furthermore, therapy instructions are counterintuitive to the typical patient’s tendency to extend time in bed in the attempt (usually futile) to gain extra sleep. Although very effective if compliance is good, the behavior therapies have obstacles to their implementation in practice, leaving scope for improvement.

## 4. Intensive Sleep Re-Training (ISR)

Intensive sleep re-training (ISR) is a recently proposed novel behavior therapy that promises to overcome difficulties of the present behavior therapies [22,23]. ISR follows principles similar to SCT, but has the advantage that the treatment period is less than 24 h. This more intensive routine eliminates the need to adhere to difficult instructions every night over a long period of time (2–4 weeks). Sleep medications are attractive to insomnia patients because of the rapid onset of symptomatic relief. ISR provides an equally rapid onset of beneficial effect without the side effects associated with hypnotic drugs, thus increasing its attraction in comparison with sleep medication [22].

ISR combines many opportunities to fall asleep quickly (i.e., 20–25 min opportunity every 30 min across a 24 h period). Shortly after the patient falls asleep during a sleep opportunity (2–3 min after Stage 1 sleep onset), they are awoken. The little amount of sleep that the patient receives is not enough to relieve sleep pressure. Therefore, sleep pressure increases across the night, which causes the patient to fall asleep more quickly. In addition, and as evident in Figure 1, the circadian rhythm of sleep propensity is also utilized by administering sleep onset trials across the maximum circadian sleepiness phase (2–9 am) [24]. This guarantees rapid sleep onsets (<5 min) for the majority of sleep attempts across the morning period, and the increasing sleep pressure continues the rapid sleep onsets for the rest of the 24 h laboratory period. 

Essentially, the experience of several weeks of SCT is condensed into a single 24 h period. In SCT, used mainly for sleep onset insomnia, the patient is likely to experience only one rapid sleep onset per night (the last sleep attempt in the night taking less than 15 min to attain sleep). If the patient adheres rigidly to SCT instructions, they will experience 14–21 rapid (less than 15 min) sleep onsets over 2–3 weeks in the context of the normal bedroom environment. This, presumably, is the experience that gradually reverses the bedroom environment from a conditioned stimulus for insomnia to a conditioned stimulus for falling asleep [1]. In the ISR protocol, sleep opportunities were administered every half hour over a 24 h period. Thus, up to 48 rapid sleep onsets could be experienced. Figure 1 shows the average time taken to fall asleep over this 24 h period for the 17 insomnia patients in the pilot study (on left) [22] and the 39 patients from the randomized control trial (right figure) [23]. 

In the pilot study, sleep latencies started at approximately 18–22 min but in the RCT study, that started about two hours later in the evening, the first average sleep latencies were shorter (14–15 min) [23]. In both studies, SOL decreased on the average over time reaching the shortest times (3–5 min) in the morning (i.e., about 2 a.m.–9 a.m.). Then, despite potentially greater circadian alertness during the rest of the day, sleep pressure from the effective sleep deprivation ensured relatively short sleep latencies (5–9 min). Thus, these patients, with mainly sleep onset insomnia reporting more than an hour on average to get to sleep in their home environment, were able to experience more than 40 rapid (<10 min) sleep onsets in this protocol.

The results of the first two clinical studies using ISR have been consistently impressive [22,23]. Following the ISR treatment, sleep diaries averaged weekly showed that sleep onset latencies were reduced by 24–30 min (38% and 44% reduction from baseline, respectively) and total sleep time increased by 34–60 min. Furthermore, daytime fatigue and other impairments were significantly reduced. In the randomized control trial, ISR was compared directly with SCT, the combination of ISR followed by SCT, and a non-treatment control group. ISR produced improvements of sleep and daytime functioning measures equal to the “gold standard” SCT condition. These beneficial results were evident in the first week following the ISR 24 h procedure but took about 3–4 weeks of SCT to reach similar levels [23]. The combination of ISR and SCT overall tended to produce better results than either treatment alone. Although the only significantly greater improvement for any one outcome measure was in the responder analysis (60% versus 40%), all outcome measures showed the same tendency for greater improvement in the combination group. These treatment gains were maintained equally well in all three treatment groups at the final follow-up time points (2 months and 6 months) in the two studies, respectively. 

ISR, as studied so far [22,23], is a laboratory-based procedure that requires the patient to remain in a sleep laboratory for 24 h. A trained polysomnographic (PSG) technician is required to monitor the PSG in real time to detect the point of sleep onset during each sleep opportunity so that the patient can be woken 2–3 min after the onset of Stage 1 sleep. The prompt detection of sleep onset and subsequent waking of the patient is necessary to maintain high homeostatic sleep drive to continue the rapid sleep onset experiences. The ISR procedures as carried out in the laboratory with a sleep technician is costly and not readily available to most insomnia patients. Therefore, although ISR is a very effective and rapid treatment for insomnia, an alternative administration method is needed to realize its benefits for the typical insomnia patient, with the goal being to translate this effective treatment to the home environment.

The two ISR studies were published in top sleep journals and have already received over 60 citations in the refereed scientific literature. The RCT published in Sleep [23] was selected by the editor for invited comments from the late Arthur Spielman, the developer of Bedtime Restriction Therapy, and Paul Glovinsky [25]. To quote their concluding paragraph.

“The findings of Harris, Lack et al. suggest that ISR approach holds great promise. Patients await a non-drug treatment for insomnia that brings relief as rapidly as medication. Clinicians in the community look forward to more widely applicable ISR-like procedures that can be implemented at home, without expensive and complicated sleep technology, by non-sleep experts. Theoreticians anticipate tests of the learning, sleep perception, and cognitive explanations of ISR’s rapid and robust benefits. This stunning demonstration by Harris, Lack, and colleagues should serve as a challenge to the field to create the next generation of theoretically driven non-pharmacological treatments for insomnia” [25] (pp. 11–12).

Harris et al. [23] was featured on the Harvard Medical School Health Publications website in 2012 [26] and concluded, “A self-administered version that could be done at home would bring it within reach of anyone with a good alarm clock, a steadfast partner, and the fortitude to sleep no more than three minutes every half hour over the course of an entire day. Let’s hope that the Australian study stimulates the creation of such home programs”.

It was also featured on the New England Journal of Medicine Journal Watch [27], which concluded, “The rapid and sustained effects of ISR make it an exciting new treatment. Clinicians should consider this option for their patients with chronic sleep-onset insomnia. Eventual development of home-based, less expensive versions of ISR should make it more widely available”.

## 5. Translation of ISR to the Home Environment

All comments suggested that ISR has an exciting potential for the rapid and effective treatment of insomnia, but a home-based version is needed to make the treatment more readily accessible. The good news is that such developments have now taken place and are under further development. An iPhone application (Sleeponq) has been created to carry out ISR in the home environment [28]. Instead of measuring PSG to determine sleep onset, it uses a behavioral response to an external stimulus. The external stimulus is a faint tone emitted every 30 s by an iPhone to which the patient must respond by moving the smartphone. The iPhone contains an in-built motion sensor that is used to recognize the response. When the patient fails to respond to the tone stimulus, the app concludes that the patient has fallen asleep and emits a loud auditory alarm to awaken the patient. After being awake for a few minutes, the patient can initiate another sleep opportunity, and so on. Therefore, assuming that these failures to respond to the tone stimuli are valid indicators of PSG sleep onset, this app should be successful in carrying out the ISR procedure without the need for PSG or sleep technicians. The app is available for purchase to anyone with an iPhone at a small expense.

So what about the question, “Are failures to respond to external stimuli valid measures of sleep?” It appears from some existing research that using behavioral responses to an auditory stimulus is a relatively accurate method of detecting sleep onset [29]. Research with the iPhone app and similar devices have shown that responses tend to cease 2–3 min after Stage 1 sleep onset. This is likely due to the auditory threshold sharply increasing after sleep onset, meaning that the likelihood of hearing low-intensity tone stimuli is reduced after sleep onset. For the purpose of administering ISR, the additional 2–3 min of light sleep that the patient receives before the app detects that they have fallen asleep would not be enough to alleviate sleep pressure, such that a high sleep drive would be maintained across the night and following day similar to the ISR laboratory studies. Therefore, it is likely that this method of detecting sleep onset is accurate enough for administering ISR in the home environment without impacting treatment efficacy. The app developers have reported considerable anecdotal support from their customers, but experimental evidence of treatment efficacy is still needed.

## 6. Future Research to Improve the Effectiveness of Intensive Sleep Re-Training

### 6.1. Distributed Versus Massed Trials

The Sleeponq app developers have also reported that, for purposes of practicality, users opt to undergo ISR for the first 2–3 h of the night and then allow themselves to sleep uninterrupted for the rest of the night. This is instead of carrying out the original ISR procedure in which all practice trials are amassed over the full 24 h. In the abbreviated method, the user may be able to experience enough rapid sleep onsets over a few nights (8–10 per night) to result in the same benefits that the original ISR procedure provides in one 24 h period. This suggestion has the attraction of a less arduous sleep deprivation period (2–3 h versus 24 h) although it may incur the need for this partial deprivation over several nights. If the number of sleep onset trials is fixed (e.g., 30), does treatment efficacy differ dependent on whether the trials are distributed or massed (i.e., one rapid sleep onset per night as in traditional stimulus control therapy, 7–8 sleep onsets/night over 4 nights as suggested by the iPhone app, or 30 sleep onsets over one complete night of sleep deprivation)? The therapeutic efficacy of distributed versus massed trials still needs to be experimentally investigated. If there is little difference between these alternatives, the chosen option may then be determined more by practical considerations and the desired speed of improvement. 

### 6.2. Robust Recovery Sleep

There are some practical advantages and perhaps some therapeutic elements of the original ISR procedure that may favor it over the model of distributed treatment over several nights. In the original ISR protocol, the treatment can be confined to one 24 h period on the weekend (e.g., Friday night and Saturday or Saturday night and Sunday as was used in the ISR studies), thus avoiding conflict with work/school commitments. The participants in both of these studies returned home and experienced robust and full recovery sleep on the Sunday night and were able to meet commitments the following week with no difficulty [23]. Many of them remarked about the satisfying and lengthy recovery sleep they experienced and felt reassured that they still had a robust sleep capacity, as was evident in their recovery sleep. Given many insomnia patients believe their sleep mechanism has been irreparably damaged by their chronic insomnia, ISR also serves as a behavioral experiment to dispel this maladaptive belief. This is an effective cognitive therapy element of ISR, at least when receiving recovery sleep after sleep deprivation, as in the massed practice of the original ISR procedure.

### 6.3. The Experience of Sleep Deprivation

Another potential cognitive therapy element of the original ISR protocol is the experience of total sleep deprivation. The typical daytime feeling reported by those with chronic insomnia is not sleepiness but fatigue [30]. It is the frequent feelings of fatigue that are burdensome and potentially provide the main motivation for seeking treatment [2]. Feelings of sleepiness, on the other hand, are rare but desired in the insomnia population. The overwhelming feelings of sleepiness experienced by the participants in the ISR studies [22,23] were novel for most of the patients. Patients found it reassuring that they could again feel sleepy and fall asleep quickly under the sleep deprivation conditions. They also found it reassuring that sleep loss, something they typically tried to avoid by giving themselves too much time in bed, did not have the catastrophic effects they feared. It only made them extremely sleepy—a desirable outcome that they rarely experienced. Their sleepiness was evident by their struggle to remain awake between the half-hourly trials in the ISR protocol and needed the assistance of conversations with a researcher in order to remain awake. These unexpected positive effects of the ISR procedure possibly contributed to the unanimous compliance with the 24 h sleep deprivation and awakening procedure. Of the 56 patients in the two studies implementing the ISR protocol, all of them completed the full protocol despite ethically directed instructions that they were free to leave the study at any time. This outcome compares very favorably with experimental clinical trials of CBT-I in general [21]. Therefore, despite trepidation of many patients prior to the study, the patients treated with ISR did not find it an aversive experience. If both the experience of total sleep deprivation and robust recovery sleep in the original ISR protocol are helpful cognitive therapeutic elements, it would tend to favor the massed practice protocol, but this needs empirical confirmation.

### 6.4. Feedback of Time Taken to Fall Asleep

Receiving feedback on every trial about the length of time taken to fall asleep could also be a therapeutic element of ISR. Since awakenings occurred only 2–4 min from the start of Stage 1 sleep, participants were awoken in most cases out of Stage 1 or light Stage 2 sleep. Awakenings from this light sleep are often not perceived as awakenings from sleep but as continued wakefulness, especially for those with chronic insomnia [31]. Those not perceiving an awakening from sleep would conclude they had not yet fallen asleep. Getting feedback about the length of time taken to objectively fall asleep should help to improve their sleep/wake state perception. Those suffering chronic insomnia typically underestimate their total sleep time by 1–2 h [17,32]. Thus, improving sleep/wake state perception should, on its own, improve the subjective estimates of total sleep time, reduce anxiety about sleep, and improve daytime symptoms [33]. Given that the ability to fall asleep is a learned skill, feedback about their improving performance should enhance that skill [34,35]. Providing immediate feedback of time taken to fall asleep on every trial requires some programming resources but may not be necessary. Feedback at the end of a training session may be equally beneficial. This question needs investigation. 

### 6.5. Effectiveness of ISR on Different Insomnia Phenotypes

The two ISR studies [22,23] selected patients predominantly with sleep onset insomnia who, nevertheless, also had substantial amounts of wake time after sleep onset (WASO) (e.g., 70–80 min). Following ISR alone, sleep latency was reduced, but WASO did not differ from the controls. WASO was reduced when ISR was combined with SCT, which provides a degree of ongoing sleep restriction in patients. This suggests that ISR alone may not be effective in treating those with sleep maintenance or early morning awakening insomnia predominantly. Therefore, the comparison of the efficacy of ISR alone and combined with other ongoing CBT-I elements as well as the standard CBT-I package needs to be made with different insomnia phenotypes. 

## 7. Potential Improvements of the Behavioral Response Measure of Sleep Onset

The iPhone app (Sleeponq) uses the failure to respond to an external stimulus as the measure of sleep onset. The existing research suggests this is relatively accurate to PSG sleep onset (2–3 min later than PSG Stage 1 onset on average) [28] and most probably accurate enough to carry out ISR effectively. However, it would appear that some improvements can be made to the response measure of sleep onset. One potential problem is the use of the auditory modality for the external stimulus. It is presumed that the lowest perceptible tone stimulus should be used to avoid being awoken by the tone stimuli of a higher intensity, thus never failing to respond and achieving sleep according to the app. However, ensuring that the tone intensity is calibrated to the lowest perceptible intensity and then is not affected by the position and proximity of the iPhone to the user’s ear can be problematic. This could be somewhat overcome by the use of an ear bud to remove the factor of iPhone proximity, but maintenance of ear bud placement can be unreliable.

The other potential improvement in this sleep measure is the choice of response. Ideally, the minimal voluntary response detectible would be the least intrusive and most sensitive behavioral response for this purpose. The present iPhone app requires the iPhone to be held in the hand and be moved with a back-and-forth motion of the wrist. This degree of effort and motor activity, although small, may still contribute to unnecessarily prolonged wakefulness. This amount of motor effort can certainly be reduced with the choice of another response. Such a response could be a small wiggle of one finger. A small, lightweight wearable device placed on one finger like a thimble or ring could contain a motion sensor activated by a small finger wiggle. This wearable finger device could also include a vibratory stimulus to act as the external stimulus to which the user responds. Such a device would obviate the problems associated with a less controllable auditory tone stimulus or the need to wear ear buds while trying to fall asleep.

Another advantage using a small finger device is that it could free the user from the need to sleep with the smartphone. Although some smartphone apps related to sleep require the phone to be located either in the bed with the sleeper or next to the bed, many people would prefer not having the phone in or next to the bed. A wearable device could be programmed with a smartphone but then be able to run independently of the smartphone placed outside of the bed or bedroom. 

## 8. Development of THIM Wearable Device for Sleep

A device as suggested above is now in development with the help of crowdfunding support from Kickstarter at [36]. It will be programmable with a smartphone but then run independent of the smartphone in carrying out several sleep-related functions. It will make use of the external stimulus response indicator of sleep/wake state to determine sleep onset rather than the less accurate passive movement indicator of sleep/wake as in actigraphy [37]. Since it will accurately identify the time of sleep onset without the need for PSG recording methods, it will be able to carry out the awakening procedures required in the administration of ISR outside of the sleep laboratory in the insomnia patient’s own home environment. Not only will it make this very effective and rapid insomnia treatment more readily available in the user’s home environment, it could very well be more effective than the published results from the laboratory studies. Because the most effective treatment would be to reverse the association of the alerting insomnia response with the bed environment of the sufferer, the re-training trials should be most effectively done in the insomnia sufferer’s normal bed environment. Thus, ISR would be more accessible in the home environment to all insomnia sufferers and may well be more effective than the already promising laboratory results.

## 9. Other Future Research Projects

Will the use of a tactile stimulus and finger movement response with the Thim device be as accurate or more accurate an indicator of EEG sleep onset and PSG determined sleep/wake state than a hand movement response to an external auditory stimulus? How will this accuracy compare to the passive movement indicator of the sleep/wake state in actigraphy? Will the answers to these questions be the same in those with chronic insomnia sufferers as in good sleepers?Can a Thim type device effectively administer ISR and obtain effective treatment outcomes?If ISR is administered in only one long session (e.g., across one entire night or night and following day), how many sleeping trials are required to reach maximal or near maximal improvement? The earlier versions of ISR in the laboratory spaced the sleep attempt trials apart by 30 min. This was not a choice based on treatment effectiveness but only on administrative convenience when running two or more patients simultaneously in the laboratory protocol. If trials do not have to delay until each half-hour time point, subsequent trials can be run as soon as the participant has awoken from the previous trial, thus increasing the frequency of trials across the night. In this way, a participant may be able to complete up to 40 quick sleep onsets within 8 h and have the rest of the following day available for other activities, keeping in mind the sleepiness inducing effect of the virtual total sleep deprivation and avoiding potentially dangerous situations (e.g., driving while excessively sleepy). A study varying the number of sleep onset trials could determine in general the number of trials necessary to reach near maximal treatment effectiveness. It may be considerably less than 40 trials.Should the break between sleep onset trials in ISR simulate the normal process of going to bed each night? The conditioning model of chronic insomnia would suggest that many of the cues or elements of deciding to go to bed and the process of doing so may have developed as conditioned stimuli eliciting an arousal (fight or flight) insomnia response. These processes should be reversed by replicating them each time preceding a rapid sleep onset. However, this would require getting out of bed after being awoken by the ISR program and going back to the normal location when deciding to go to bed (e.g., TV room or computer room) and resuming those pre-bed activities for a few minutes before returning to the bedroom and initiating another sleep onset trial. Although this procedure may provide somewhat more effective re-association of all the possible conditioned stimuli from being triggers for arousal to becoming triggers for sleepiness, it may be arduous for some individuals, especially in the colder winter months, and likely to receive poor adherence. It would also slow down the rate of sleep onset trials. Another option would be to initiate another sleep onset trial immediately after being awoken on the previous trial. This would be much easier and probably have good adherence. It would also allow many more sleep onset trials in a given period of re-training. Although each trial may be individually less effective than when following the longer procedure, it may be compensated by more trials administered in a given time. Research is needed to determine which of these two methods overall is a more effective insomnia treatment.

## 10. Conclusions

Intensive sleep re-training (ISR) has strong empirical support from laboratory studies as a very effective treatment of chronic insomnia. Its initial main disadvantage was the expense and inaccessibility of an intensive laboratory procedure. That disadvantage appears to have been largely overcome by the development of an iPhone app, Sleeponq. It uses the failure to respond to an external stimulus as the indication of sleep onset and uses algorithms from the iPhone to administer the protocol of ISR, thus obviating the need for PSG recording and laboratory technicians. However, further improvements can be made with the use of a minimal finger movement response to a miniscule vibratory stimulus built into a small wearable device autonomous of the smartphone, such as the THIM device. This small wearable device would also facilitate research to further improve the effectiveness of this already promising ISR treatment of chronic insomnia. 

## Figures and Tables

**Figure 1 brainsci-07-00033-f001:**
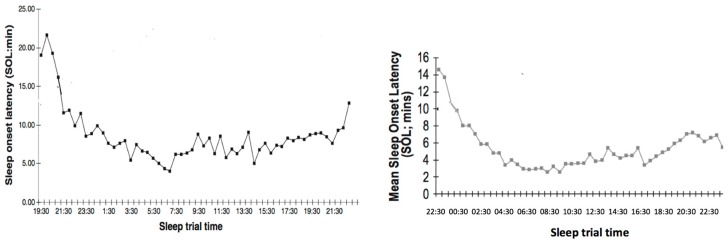
Average time taken to fall asleep over 24 h of intensive sleep re-training (ISR) during a pilot study (*n* = 17; **left**) and during a randomized control trial (*n* = 39; **right**).

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
