# Peer review of "Intensive Sleep Re-Training: From Bench to Bedside"

_brainsci, 2017, doi:10.3390/brainsci7040033_

Round 1

Reviewer 1 Report

Notes for Brain Sciences Review Paper (MS# brainsci-183596)

Overall

The paper is well written, well organized, and very informative about ISR and its future prospects ... We are very convinced that the lab based protocol holds great promise. This said, the article adopts a number of conventions we have not seen used in a journal review incl.: reporting patient anecdotes; the inclusion of expert "testimonials"; citation counts as evidence of how influential the approach has been; the presentation of new versions of the protocol (using alternative methodology) without supporting RCT data; and the inclusion of text re: potential mechanisms of action for ISR without clearly labeling such information as speculation. While all of this information was informative, it is nonetheless unconventional enough that we checked in with the journal’s editor re: the inclusion of these elements… We were assured that each of these features of the article were acceptable. Accordingly, we only have a few comments.

Strengths

l A strength of the paper is that reasons are provided for why instructions in CBT-I are difficult to follow for patients

l The use of a wearable device (THIM) to awaken patients is noteworthy considering that this is the one sense modality that is not “dysfaciltated” during sleep and thus may serve as an ideal way to awaken patients.

Limitations

l  The ISR studies by and large focused on initial insomnia, and while the results are impressive, the effects of ISR on various types and subtypes of insomnia should be explored as well.

Specific Comments

Introduction

l  It seems a bit overstated to ascribe no therapeutic effects to OTCs for insomnia (“After the extinction of placebo effects…”)

l  Suggesting that hypnotics provide only moderate symptomatic relief is a “double edged sword” type comment. We say this because the acute effects of CBT-I and BZRAs (if not other indicated meds) are similar if not identical.   

l  It may add to the paper to give DSM-5 definition for CI which includes 3 days or more per week of bad sleep for > 3 months.

l Please end the introduction with a statement about the organization of the paper…

Cognitive/Behavioral Therapy for Insomnia (CBT-I)

l It’s true that pure stimulus control instructions have #1 as a feature, but in the context of CBT-I, time to bed is determined.

l According to “Orthodox” Spielman (PMID:3563247), the rule is 15 minutes for titration in sleep restriction therapy, not 30 minutes as stated in line 68.

Effectiveness of Behavior Therapies

l We suggest that the phrase “light sleep” be replaced with the PSG criteria for awakening during each sleep opportunity.

l For figure 1, we are wondering if the author should comment on the fact that in both cases PSG SL initial values are within normal limits.

l It would be helpful if the authors clarified how long sleep opportunity lasted in each ISR study (i.e., 25 minutes [pilot] and 30 minutes [RCT]).

l Could the authors clarify on what proportion of occasions subjects were awake for the whole duration of the various sleep opportunities?

l For the sentence (line 135), please provide information regarding how many days the sleep diaries were administered.

l Please state, in addition to the average minute reductions in SL and WASO, the percent change (e.g., 20 minute reduction in SL constituted a 50% decrease in average time to fall asleep).

l For lines 142-143, please restate to say that treatment gains were maintained rather than “beneficial effects” and elaborate on how well the effects of ISR were maintained.  Further, a brief explanation of how much more effective the combination of ISR and SCT were as compared to other conditions would useful.

Translation of ISR to the home environment

l Please state what the inter-trial interval was (how many minutes the patient stays awake before the next trial).

l It appears somewhat contradictory that the last paragraph on page 4 is implying that research has yet to be conducted on validating responses to an external auditory stimulus whereas the following paragraph offers data to support for it.

Future research to improve the effectiveness of ISR

l  Please consider suggesting a comparison between ISR and proper CBT-I

Robust recovery sleep - The experience of sleep deprivation

l  While we understand the media might have mischaracterized ISR as “torture”, we recommend leaving the accusations of fake news to Donald Trump.J

Author Response

Reply to Reviewer 1

“Overall”

We thank the reviewer for the positive comments about the article overall. We also appreciate that it pushed the boundaries of convention for the sake of presenting an interesting and informative article, one that we hope will stimulate further interest in the translation of ISR to a more accessible format while hopefully retaining its efficacy. This is a daunting task, one to which our group will contribute but does not wish to monopolize. Therefore, a main purpose of this article was to stimulate other groups to this research. With this said, we will be sensitive to your comments in making revisions to the article.

“Strengths”

We were unaware that tactile sensation is “dysfacilitated” during sleep presumably meaning that responding to even low intensity tactile stimuli during EEG defined sleep is likely. We were assuming, as in other modalities (e.g. auditory) that response thresholds are raised sharply at sleep onset (or in the first 2-3 minutes into S1 sleep) such that low intensity stimuli from any modality are likely to fail to elicit a response but that very intense stimuli from the same modality even later in the descent into deeper sleep will exceed that threshold and awaken the user. If the reviewer is aware of perceptual studies that show differences in this respect for the tactile modality we would appreciate the references.

“Limitations”

Does ISR effectively treat other insomnia subtypes?

Excellent question that needs mentioning.

We have added an appropriate short paragraph in the section, “Future research to improve the effectiveness of Intensive Sleep Re-training”

“Specific Comments”

It seems a bit overstated to ascribe no therapeutic effects to OTCs for insomnia (“After the extinction of placebo effects…”)

We have deleted the sentence about placebo effects and rephrased that paragraph.

Suggesting that hypnotics provide only moderate symptomatic relief is a “double edged sword” type comment. We say this because the acute effects of CBT-I and BZRAs (if not other indicated meds) are similar if not identical.

Point well taken. We have deleted “moderate” making the emphasis on the temporary symptomatic relief.

It may add to the paper to give DSM-5 definition for CI which includes 3 days or more per week of bad sleep for > 3 months

We prefer to be descriptive rather than definitional in this article but have added reference to the duration definition of > 3 months.

Please end the introduction with a statement about the organization of the paper…

Good idea. We have added a short paragraph at the end of the introduction that we feel enhances the interest and readability of the article.

It’s true that pure stimulus control instructions have #1 as a feature, but in the context of CBT-I, time to bed is determined.

This is an interesting comment. We were not quite sure what is meant by “in the context of CBT-I time to bed is determined”. Does this assume that SRT takes precedence over SCT in the CBT-I package? Although we admit that SRT is probably used more than SCT in the typical application of CBT-I, if sleep onset is the main problem then SCT should probably take precedence. Even if SRT is mainly used, it is probably helpful to add #1 of SCT to the instructions at least for the first few nights or week until the sleep restriction and sleep drive kick in to reduce SOL to a reasonable level. In fact, adding #1 of SCT for the first few nights will probably speed up the accumulation of sleep drive and therapeutic response.

In any case we did not think any amendments to text were appropriate.

According to “Orthodox” Spielman (PMID:3563247), the rule is 15 minutes for titration in sleep restriction therapy, not 30 minutes as stated in line 68.

You are correct. In practice in the context of the rubberiness of sleep diary estimates and to expedite the schedule of treatment, we use 30 minute increments in SRT. But 15 min increments does emphasize the extended time scale of classical SRT so we have made the change in text from 30 min to 15 min. Thanks.

We suggest that the phrase “light sleep” be replaced with the PSG criteria for awakening during each sleep opportunity.

That section on page 3 has been appropriately modified.

For figure 1, we are wondering if the author should comment on the fact that in both cases PSG SL initial values are within normal limits.

Thank you for the comment. Your observation is correct. However, it is the case that PSG sleep latencies of insomnia patients is found to be within normal limits across the day (e.g. Bonnet and Arand, 1998-2000). Thus, the SOLs at the beginning of the protocols from the two ISR studies are not unexpectedly short despite their reported typical diary SOLs. But we would rather not enter such a discussion in the article.

It would be helpful if the authors clarified how long sleep opportunity lasted in each ISR study (i.e., 25 minutes [pilot] and 30 minutes [RCT]).

Good point. This information is added.

Could the authors clarify on what proportion of occasions subjects were awake for the whole duration of the various sleep opportunities?

We have those figures for the 2012 study that allowed 20 minute opportunities. In the first trial 37% of participants failed to fall asleep within the 20 minute opportunity. That percentage fell rapidly such that by Trial 5 less than 5% of participants failed to fall asleep and for the majority of trials across the 24 hours all participants fell asleep within the 20 minute opportunity.

However, we feel this does not add appreciable information to the results as described already and are reluctant to expand the ISR section further.

For the sentence (line 135), please provide information regarding how many days the sleep diaries were administered.

The phrase, “averaged weekly” was added to give this information.

Please state, in addition to the average minute reductions in SL and WASO, the percent change (e.g., 20 minute reduction in SL constituted a 50% decrease in average time to fall asleep).

Helpful point. Information added.

For lines 142-143, please restate to say that treatment gains were maintained rather than “beneficial effects” and elaborate on how well the effects of ISR were maintained.  Further, a brief explanation of how much more effective the combination of ISR and SCT were as compared to other conditions would useful.

This additional description is added.

Please state what the inter-trial interval was (how many minutes the patient stays awake before the next trial).

In the middle of that translation section we state, “After being awake for a few minutes, the patient can initiate another sleep opportunity, and so on.” This implies that in practice the sleeponq program is probably run with only a few minutes between trials. But the question of testing the effectiveness of different inter-trial intervals in ISR is elaborated in “Other future research projects #4.

It appears somewhat contradictory that the last paragraph on page 4 is implying that research has yet to be conducted on validating responses to an external auditory stimulus whereas the following paragraph offers data to support for it.

We have changed the word “if” to “assuming” to make it less contentious.

Please consider suggesting a comparison between ISR and proper CBT-I

We have included this suggestion in the added short paragraph testing ISR on different insomnia phenotypes.

While we understand the media might have mischaracterized ISR as “torture”, we recommend leaving the accusations of fake news to Donald Trump.J

We liked your reference to fake news! Admittedly the use of the term “torture” was extravagant and probably unnecessary. We received considerable publicity during the conduct of the study at a time when many Australians were trying to secure the release of an Australian detained at Quantanamo who had been subjected to sleep deprivation torture.

In any case we are comfortable in deleting this phrase.

Reviewer 2 Report

This manuscript is a review of the literature regarding non-pharmacological therapies for insomnia, with a specific focus on CBT-I and Intensive Sleep Re-training (ISR), and a discussion of novel ways to implement ISR in the home environment.  This manuscript adds to the literature by comparing and contrasting ISR (a new therapy for insomnia) to CBT-I  (a well-established therapy for insomnia) and identifying possible future research directions for ISR. Specific comments are outlined below:

1. As a review article, there are areas within the manuscript that may benefit from more citations of the existing literature:

                -Page 1, last sentence of the introduction: citations to support side effects, risk of dependency, limited duration of action of hypnotics

                -Page 2, lines 57-58 states that Stimulus Control (SCT) is “mostly used when initial sleep onset problems are predominant”—please provide a citation for this statement.

                -Similarly, page 2, lines 59-60 describe Sleep Restriction Therapy (SRT) as being “most commonly indicated” for sleep maintenance insomnia— please provide a citation for this statement.

                -I believe there are more citations for side effects/typical compliance with CBT-I/SRT/SCT that could be added to the section of page 2, lines 83-94.

                -A citation for the statement on page 6, line 271 would be helpful (“the existing research suggests this is relatively accurate to PSG sleep onset”).

2. This section (“Cognitive/Behavior Therapy for Insomnia (CBT-I)” is a bit confusing as written. It seems that the focus of this section is to both review CBT-I as a treatment package and the components of CBT-I that have evidence as stand-alone therapies. Perhaps it could be structured such that CBT-I as a multi-component therapy is described in the first paragraph, with citations for its evidence as a treatment package, then followed by paragraphs outlining the specific behavioral therapies that have evidence to support their use as single-component treatments for insomnia (e.g., SCT and SRT).  

 3. The 3rd section of the manuscript (“Effectiveness of Behavior Therapies”) focuses more on the barriers/challenges to implementing the therapies (and less on data regarding their effectiveness)—the section heading could be amended to better reflect its content.

4. Page 3, line 114, the sentence “In SCT the patient experiences only one rapid sleep onset per night” (and the sentence after) are likely inaccurate and citations should be provided for them. Depending on how many times the patient awakens, they may experience more than 1 rapid sleep onset per night.  I am not aware of any studies that have quantified the average number of rapid sleep onsets experienced in a typical night of SCT.  

Author Response

We thank the reviewer for the thoughtful precis of the article and time taken to make helpful suggestions that will improve the thoroughness of the review and its impact on readership.

Page 1, last sentence of the introduction: citations to support side effects, risk of dependency, limited duration of action of hypnotics

Two relevant references have been added.

-Page 2, lines 57-58 states that Stimulus Control (SCT) is “mostly used when initial sleep onset problems are predominant”—please provide a citation for this statement.

And -Similarly, page 2, lines 59-60 describe Sleep Restriction Therapy (SRT) as being “most commonly indicated” for sleep maintenance insomnia— please provide a citation for this statement.

Without being able to find explicit references from the literature for these statements, we have changed the wording to be less authoritative. This shows the need for more work on tailoring treatment to insomnia phenotype. Would a variety of insomnia types randomly allocated to either SCT or SRT show differences in effectiveness dependent on types? We are unaware of such a study. Clearly needs to be done.

-I believe there are more citations for side effects/typical compliance with CBT-I/SRT/SCT that could be added to the section of page 2, lines 83-94.

Two new references have been added.

A citation for the statement on page 6, line 271 would be helpful (“the existing research suggests this is relatively accurate to PSG sleep onset”).

Reference added.

2. This section (“Cognitive/Behavior Therapy for Insomnia (CBT-I)” is a bit confusing as written. It seems that the focus of this section is to both review CBT-I as a treatment package and the components of CBT-I that have evidence as stand-alone therapies. Perhaps it could be structured such that CBT-I as a multi-component therapy is described in the first paragraph, with citations for its evidence as a treatment package, then followed by paragraphs outlining the specific behavioral therapies that have evidence to support their use as single-component treatments for insomnia (e.g., SCT and SRT).

We have accommodated this good suggestion by enlarging the first paragraph and starting a new paragraph with the behavior therapies.

 3. The 3rd section of the manuscript (“Effectiveness of Behavior Therapies”) focuses more on the barriers/challenges to implementing the therapies (and less on data regarding their effectiveness)—the section heading could be amended to better reflect its content.

Good point, the sub-heading has been amended to more appropriately label this section.

4. Page 3, line 114, the sentence “In SCT the patient experiences only one rapid sleep onset per night” (and the sentence after) are likely inaccurate and citations should be provided for them. Depending on how many times the patient awakens, they may experience more than 1 rapid sleep onset per night.  I am not aware of any studies that have quantified the average number of rapid sleep onsets experienced in a typical night of SCT.  

You are correct for the case of using SCT for night time awakenings, in practice there may be as many as 3-4 per night. We, also, are unaware of any studies that have actually measured the number of rapid sleep onsets per night using SCT. There are studies that documented the number of times out of bed before attaining sleep rapidly. Usually this shows perhaps 3-5 times out of bed in the first few nights and a gradual decrease to 1-2 over the weeks of therapy.

Our working assumption was that, in practice, SCT is used mostly for sleep onset insomnia and that after finally attaining first rapid sleep onset for the night, which in the first few nights of treatment instructions may be quite late, there would be few if any awakenings until reaching the forced early morning awakening using the alarm clock.

In any case we have amended the wording of that second sentence of the paragraph to qualify the statement and make it less contentious.